# Effects of stigma on help-seeking behavior in mental health: A community-based study in Ghana's Sekyere South District in the Ashanti region

## Research Article

cross-sectional studies; Ghana; help-seeking behavior; health stigma and discrimination framework; mental illness-related stigma

**Corresponding author:**
Emmanuel Kwasi Afriyie;
Email: afriyiekwasi@gmail.com

E.K.A. and E.K.N.B. authors contributed equally to this article.

Emmanuel Kwasi Afriyie[1,2] (ORCID), Emmanuel Kofi Nti Brantuo[3],
Samuel Egyakwa Ankomah[4], Emmanuel Kumah[5], Godfred Otchere[6],
Precious Wonder Adekore[4] and Joseph Atta Amankwah[7,8]

[1]Komfo Anokye Teaching Hospital, Ghana; [2]Southern Medical University, China; [3]Department of Public Health, Catholic University College of Ghana, Ghana; [4]Department of Management, University of Cape Coast, Ghana; [5]Department of Health Administration and Education, University of Education Winneba, Ghana; [6]Department of Preventive and Social Medicine, University of Otago, New Zealand; [7]School of Public Health, College of Health Sciences, KNUST, Ghana and [8]General Administration, Methodist Faith Healing Hospital, Ghana

## Abstract

Mental illness-related stigma acts as a critical barrier to care by fostering shame, fear of judgment and discrimination, which deters individuals from seeking help, delays treatment and worsens outcomes. This study aimed to investigate the forms, drivers and consequences of mental health-related stigma on help-seeking behavior. A cross-sectional study was conducted from November 2020 to March 2021. Data were collected from 419 participants using structured questionnaires, guided by the Health Stigma and Discrimination Framework and the Attitudes Towards Seeking Professional Psychological Help Scale for validation. Data were analyzed using Statistical Package for the Social Sciences version 22, employing descriptive statistics, chi-square tests and multinomial logistic regression. The average age of participants was 34.5 years. Findings revealed alarmingly high stigma: economic (76.8–80.2%), social (77.1–81.2%) and psychological (71.9–82.8%). Key drivers included stereotypes of dangerousness (58.7%) and systemic healthcare discrimination (65.6%). Multinomial regression confirmed that all stigma forms significantly reduced help-seeking odds. Structural barriers (odds ratio [OR] = 0.48) and internalized shame (OR = 0.53) were the strongest deterrents, with a multiplicative effect for combined economic and psychological stigma (OR = 0.41). This complex, multilayered barrier, driven by socio-cultural beliefs and structural failures, necessitates urgent, multifaceted interventions targeting public education, policy and self-stigma to improve mental health equity in rural Ghana.

## Impact statements

This research provides critical, evidence-based insights into the complex nature of mental health stigma in a rural Ghanaian district, revealing it as a multilayered barrier driven by socio-cultural beliefs and structural failures. Our findings move beyond urban-centric data to show that stigma significantly reduces help-seeking, with economic exclusion and internalized shame being the strongest deterrents. This study offers a vital blueprint for policymakers and health advocates, demonstrating that effective anti-stigma campaigns must be multifaceted, simultaneously targeting public education, institutional policies and individual self-stigma. The evidence is crucial for designing targeted interventions to improve mental health equity in rural Ghana and similar low-resource settings globally.

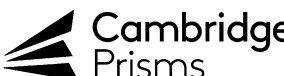



## Introduction

Mental illness-related stigma presents a significant global barrier to care, leading to delayed treatment, worsened symptoms, social exclusion and poor health outcomes (Sickel et al., 2014; Ssebunnya et al., 2019; Yu et al., 2023). This stigma stems from widespread misunderstanding and prejudice, causing individuals to fear being labeled, ostracized and/or judged. These fears diminish self-esteem and create a profound reluctance toward help-seeking for mental health illness (Thornicroft et al., 2022; Girma et al., 2024). With ~450 million people affected by mental illness worldwide (World Health Organization., 2022), addressing this barrier is a critical public

health concern. Previous studies highlight that stigma operates at multiple, interconnected levels, creating complex barriers to care (Chukwuma et al., 2024; Thi et al., 2024).

Public stigma involves the negative attitudes and discriminatory behaviors held by the wider society (Liamputtong and Rice, 2021; Thornicroft et al., 2022). Self-stigma occurs when individuals internalize these negative beliefs, leading to shame and a loss of self-efficacy. Structural stigma refers to discriminatory policies and practices within institutions, such as healthcare systems and workplaces (Liamputtong and Rice, 2021; Thornicroft et al., 2022). Together, these forms of stigma can lead to social isolation, internalized shame and inadequate access to services, which severely impact treatment-seeking behaviors and overall health outcomes (Prizeman et al., 2023). While these dynamics are increasingly documented in high-income countries, their interplay in low-resource settings remains underexplored (Thornicroft et al., 2022).

In Ghana, for instance, ~10.7% of the population lives with a mental health condition (Ae-Ngibise et al., 2017; Metrics and Evaluation, 2024). However, most are reluctant to access mental health care (Harden et al., 2023). Existing research in Ghana has primarily focused on urban areas, neglecting rural districts where traditional beliefs and limited mental health infrastructure intensify stigma (Amadu and Hoedoafia, 2024; Daliri et al., 2024). This leaves a significant gap in understanding stigma-related factors and how they affect mental help-seeking, particularly in rural communities such as the Sekyere South District of Ghana. Although Ghana's Mental Health Act (846 of 2012) and global calls for equity highlight the urgency of addressing stigma, a clear understanding of its context-specific manifestations in rural areas is lacking (Walker and Osei, 2017; Moitra et al., 2023).

This study addresses these gaps by examining the forms, drivers and consequences of mental illness-related stigma in the Sekyere South District, a rural region in Ghana. Using a community-based approach, we assessed how stigma operates across social, economic and healthcare systems. By incorporating perspectives from patients, caregivers and healthcare workers, this research provides novel insights into the localized barriers to help-seeking for mental health illness.

## Materials and methods

### Study design and setting

Our study employed a descriptive cross-sectional design using quantitative methods. The design was selected as it facilitated analysis of stigma-related factors, as well as examination of associations between variables (e.g., stigma and help-seeking behavior) at a single point in time (Olsen and St George, 2004; Kesmodel, 2018). The study was conducted in the Sekyere South District of the Ashanti Region, Ghana. The district's administrative capital is Agona Ashanti, located 37 km from Kumasi. Data were collected from November 2020 to March 2021 across four randomly selected health facilities: Agona Government Hospital, Kona Health Centre, Jamasi Health Centre and Wiamoase SDA Hospital. These facilities were chosen to represent the diverse subdistricts within the broader study area.

### Study population, participant profile and sampling

The study population comprised three distinct groups, each chosen for their unique perspective on mental health stigma and help-seeking: (1) Patients: Adults (≥18 years) diagnosed with a mental health condition and actively receiving treatment. They were

included to provide first-hand accounts of experienced stigma and its direct impact on their help-seeking journey; (2) Caregivers: Adults (≥18 years) caring for a patient with mental illness were selected as they often witness and are affected by stigma, which can influence their support for the patient's treatment adherence and future help-seeking; and (3) Healthcare workers with at least 2 years of experience in mental health services. They were included to offer an institutional and professional perspective on stigma, including their observations of public attitudes and their own potential implicit biases.

Using Yamane's formula $n = N/1 + N(e)^2$ for a finite population (Yamane, 1973), where $n$ = sample size, $N$ = population size and $e$ = margin of error. A sample size of 399 was calculated at a 95% confidence level and a 5% margin of error. This was increased to 419 to account for a potential 5% nonresponse rate.

A two-stage sampling technique was used. First, four health facilities were randomly selected from a pool of 12 in the district using a lottery method. Subsequently, a purposive sampling technique was employed within each selected facility to recruit the specific categories of participants. This nonprobability approach was necessary and appropriate to deliberately reach the specialized, information-rich groups central to the research question. Eligible participants were identified in collaboration with the mental health units at each facility.

### Data collection instrument and procedure

Data were collected using a structured questionnaire, developed by integrating two key frameworks: (1) the Health Stigma and Discrimination Framework and (2) the Attitudes Towards Seeking Professional Psychological Help (ATSPPH) Scale (Surgenor, 1985; Stangl et al., 2019). The Health Stigma and Discrimination Framework guided the development of questions on the drivers, impacts and outcomes of stigma related to mental health conditions. The ATSPPH scale is a validated tool adopted to measure help-seeking attitudes. The instrument was divided into sections capturing sociodemographic characteristics, forms of stigma (measured on a 5-point Likert scale from "*Not at all*" to "*Very often*"), drivers of stigma (assessed via eight items on a 5-point Likert agreement scale) and help-seeking behavior using the validated ATSPPH scale (see Supplementary Material 1: Questionnaires). To ensure comprehension and cultural appropriateness, the instrument was translated into Twi and back-translated into English. It was pretested in a comparable district, Mamponteng, and refined for clarity and relevance. Trained research assistants administered the questionnaires in either English or Twi based on participant preference.

### Validity and reliability

The validity and reliability of the instrument were strengthened through multiple strategies. Content validity was established by grounding the questionnaire in established theoretical frameworks. The reliability of the help-seeking construct was supported by the use of the psychometrically validated ATSPPH scale, and pretesting of the questionnaire confirmed the internal consistency of the adapted instrument in the local context. Furthermore, by collecting data from three distinct participant groups involving patients, caregivers and healthcare workers, the study incorporated multiple perspectives, thereby enriching the analysis of mental health stigma within the same socio-ecological system.

### Data analysis

Data were analyzed using the Statistical Package for the Social Sciences version 22. Descriptive statistics (frequencies, percentages, means and standard deviations) were used to summarize socio-demographic characteristics and the prevalence of stigma. Inferential statistics were applied to examine relationships; specifically, chi-square tests assessed associations between forms of stigma and sociodemographic variables, and multinomial logistic regression was employed to evaluate the impact of stigma on help-seeking attitudes while adjusting for potential confounding covariates. Statistical significance was set at a $p$-value < 0.05.

### Ethical considerations

Ethical approval was obtained from the Ghana Health Service Ethics Review Committee (GHS-ERC 040/06/21). Before data collection, all participants provided written informed consent after the study purpose, procedures, risks and benefits were explained. Participants were assured of the confidentiality of their responses, the voluntary nature of their participation and their right to withdraw at any time during the data collection process without any penalty or consequence to their care or employment. However, none of the participants withdrew after providing informed consent.

### Results

#### Characteristics of the participants

The study involved 419 participants from Sekyere South District, Ghana, including mental health patients, their caregivers and healthcare workers (Table 1). The sample comprised 235 females (56.0%) and 184 males (43.9%), with most (52%) aged between 20 and 35 years. Participants under 20 years and over 65 years made up smaller proportions (9.3% and 3.6%, respectively). Religious affiliation showed that 79.0% identified as Christian. Concerning marital status, 46.3% were married and 39.4% were single, with fewer being divorced (5.3%), widowed (6.4%) or separated (2.6%). Education levels varied, with 31.3% having attained tertiary education and only 9.3% having no formal education. Regarding occupation, the majority (49.9%) were traders – engaged in small-scale commerce, usually in local markets or from small storefronts – highlighting the large informal sector in the Ghanaian economy. This was followed by healthcare workers (20.8%), students (16.7%) and business owners (12.6%).

#### Forms of stigma on mental health clients

The study examined stigma experiences across four key dimensions involving economic, social, psychological and structural, revealing distinct patterns in prevalence and intensity. The findings demonstrate widespread stigmatization, though with notable variations across different life domains. Economic stigma emerged as particularly pervasive, with financial exclusion being the most frequently reported concern (80.2%). Workplace-related stigma manifested strongly, as 78.6% experienced employment discrimination (34.9% very often) and 76.8% reported workplace marginalization (38.1% very often). These patterns suggest systemic barriers to economic participation and professional integration. Social stigma indicators revealed alarming prevalence. Verbal abuse was the most common form (81.2%), with 44.4% experiencing it very frequently. Relational stigma was similarly widespread, with family exclusion and

**Table 1.** Summary of characteristics data of study participants (N = 419)

| Characteristics | Category | Frequency (n) | Percent (%) |
|---|---|---|---|
| Gender | Male | 184 | 43.9 |
| | Female | 235 | 56.0 |
| Age (years) | Below 20 | 39 | 9.3 |
| | 20–35 | 218 | 52.0 |
| | 35–50 | 96 | 22.9 |
| | 51–65 | 51 | 12.2 |
| | Above 65 | 15 | 3.6 |
| Religion | Christian | 331 | 79.0 |
| | Muslim | 65 | 15.5 |
| | Traditionalist | 23 | 5.5 |
| Marital status | Single | 165 | 39.4 |
| | Married | 194 | 46.3 |
| | Divorced | 22 | 5.3 |
| | Widowed | 27 | 6.4 |
| | Separated | 11 | 2.6 |
| Level of education | Primary | 42 | 10.0 |
| | JHS | 100 | 23.9 |
| | Secondary | 107 | 25.5 |
| | Tertiary | 131 | 31.3 |
| | None | 39 | 9.3 |
| Occupation | Healthcare worker | 87 | 20.8 |
| | Student | 70 | 16.7 |
| | Trader | 209 | 49.9 |
| | Businessman/woman | 53 | 12.6 |

social isolation each affecting 77.1% of participants, and 37.2–41.1% experiencing these very often. These findings underscore the profound interpersonal consequences of stigmatization. The psychological dimension showed particularly severe impacts, with internalized shame affecting the highest proportion of participants (82.8%) and nearly half (45.8%) experiencing it very often. Negative labeling, while slightly less prevalent (71.9%), remained substantial, with 31.3% encountering it very frequently. These results highlight the deep psychological toll of stigma. Structural stigma exhibited domain-specific variations. While healthcare discrimination affected 49.6% of participants (21.0% very often), educational exclusion was markedly more prevalent (77.5%), with nearly half (47.2%) experiencing it very often. This striking disparity points to potentially different institutional mechanisms operating in these sectors (Table 2).

#### Drivers of stigma

The analysis identified several key drivers perpetuating mental health stigma in the district (Table 3). Socio-cultural factors played a significant role, with 58.7% of participants endorsing dangerousness stereotypes and 42.9% believing mental illness indicated incompetence. Structural barriers were prominent, as 48.7% cited

**Table 2.** Forms of mental health-related stigma among participants (*N* = 419)

| Stigma dimension | Specific indicator | Frequency (*n*) (%) | Response distribution *n* (%) | | | |
|---|---|---|---|---|---|---|
| | | Often + Very often | Not at all | Rarely | Often | Very often |
| Economic | Employment discrimination | 329 (78.6) | 54 (12.9) | 36 (8.6) | 183 (43.7) | 146 (34.9) |
| | Financial exclusion | 336 (80.2) | 57 (13.6) | 26 (6.2) | 187 (44.6) | 149 (35.6) |
| | Workplace marginalization | 322 (76.8) | 51 (12.2) | 46 (11.0) | 162 (38.7) | 160 (38.1) |
| Social | Verbal abuse | 340 (81.2) | 48 (11.5) | 31 (7.4) | 154 (36.8) | 186 (44.4) |
| | Family exclusion | 323 (77.1) | 59 (14.1) | 37 (8.8) | 167 (39.9) | 156 (37.2) |
| | Social isolation | 323 (77.1) | 63 (15.0) | 33 (7.9) | 155 (37.0) | 168 (41.1) |
| Psychological | Internalized shame | 347 (82.8) | 49 (11.7) | 23 (5.5) | 155 (37.0) | 192 (45.8) |
| | Negative labeling | 301 (71.9) | 78 (18.6) | 40 (9.5) | 170 (40.6) | 131 (31.3) |
| Structural | Healthcare discrimination | 208 (49.6) | 151(36.0) | 60 (14.3) | 120 (28.6) | 88 (21.0) |
| | Educational exclusion | 325 (77.5) | 61(14.6) | 33 (7.9) | 127 (30.3) | 198 (47.2) |

inadequate workplace protections and 65.6% reported systemic healthcare discrimination. Misinformation contributed substantially to stigma, with 41.7% fearing casual infection transmission and 30.8% attributing mental illness to personal failure. Institutional factors further compounded the problem, as 37.5% perceived mental health patients were perceived as inferior, and 34.1% of healthcare workers reported insufficient safety protocols in their facilities.

### Attitudes toward help-seeking

Assessment of help-seeking attitudes using the ATSPPHS revealed a complex pattern of responses. On the positive side, 60.6% of participants indicated they would seek professional help during a mental health crisis, while 47.5% expressed belief in the efficacy of psychotherapy. Nearly half (46.7%) reported willingness to seek treatment for prolonged mental distress. However, countervailing negative attitudes were also prevalent, with 52.5% expressing doubts about psychological counseling, 54.9% viewing professional help as a last resort and 56.3% preferring to rely on self-management of emotional problems (Table 4).

### Mental health-related stigma and help-seeking behavior

Multinomial regression analysis revealed significant associations between various forms of stigma and help-seeking behaviors. The model demonstrated good fit ($\chi^2 = 42.36$, $p < 0.001$), explaining 31% of the variance in help-seeking intentions (Nagelkerke $R^2 = 0.31$). Participants who experienced economic stigma showed significantly reduced odds of seeking professional help (odds ratio [OR] = 0.62, 95% confidence interval [CI] = [0.51–0.76], $p$ = 0.003). Specifically, employment discrimination decreased help-seeking likelihood by 38% (OR = 0.62), while financial barriers reduced service utilization by 41% (OR = 0.59, 95% CI = [0.47–0.73], $p$ = 0.001). Social stigma components exhibited varying effects: Verbal abuse was associated with 29% lower odds of help-seeking (OR = 0.71, 95% CI = [0.58–0.87], $p$ = 0.008), while family exclusion showed the strongest negative association (OR = 0.53, 95% CI = [0.42–0.67], $p$ < 0.001). Psychological stigma demonstrated particularly robust effects, with internalized shame reducing help-seeking odds by 47% (OR = 0.53, 95% CI = [0.41–0.68], $p$ < 0.001).

Notably, structural barriers in healthcare systems showed the most pronounced negative association (OR = 0.48, 95% CI = [0.37–0.62], $p$ < 0.001), indicating that systemic discrimination more than halved the likelihood of service utilization. The model also identified significant interaction effects between stigma types, with the combination of economic and psychological stigma producing multiplicative negative effects on help-seeking (interaction OR = 0.41, 95% CI = [0.32–0.53], $p$ < 0.001). These findings demonstrate that different stigma dimensions have distinct but interrelated impacts on help-seeking behavior, with structural and psychological stigma showing particularly strong deterrent effects. The results highlight the need for multifaceted interventions addressing both individual-level stigma perceptions and systemic barriers to care (Table 5).

### Discussion

This study investigated the forms, drivers and consequences of mental health-related stigma on help-seeking behavior in rural Ghana, specifically the Sekyere South District. Findings revealed a complex and multifaceted impact of mental illness-related stigma, which both aligns with and challenges existing global and national literature by introducing novel, context-specific factors from a rural Ghanaian setting. The economic dimension of stigma was especially pervasive, with over 80% of participants reporting financial exclusion and employment discrimination, a finding consistent with past studies (Evans-Lacko et al., 2013; Thornicroft et al., 2022). Moreover, as reported in the study by Liamputtong and Rice (2021), we found that a high prevalence of social stigma, such as verbal abuse and family exclusion, reflects well-documented, deeply ingrained cultural beliefs that link mental illness with shame or moral weakness. Furthermore, our data on psychological stigma strongly support the conclusions of Yu et al. (2023), which connect self-stigma to lower help-seeking intentions (Yu et al., 2023). The high level of internalized shame reported by our participants highlights a universal truth in stigma research: Negative public attitudes often become internalized, creating a significant barrier to care. This relationship is further reinforced by evidence that self-stigma mediates the relationship between public stigma and treatment avoidance (Favina et al., 2025).

The quantified impact of stigma on help-seeking also aligns with established models. Our regression analysis, which found that

**Table 3.** Drivers of mental health-related stigma (*N* = 419)

| Stigma category | Specific driver | Endorsement rate (%) | Key findings |
|---|---|---|---|
| Socio-cultural factors | Dangerousness stereotypes | 58.7% | The majority associated mental illness with violent/dangerous behavior. |
| | Belief in incompetence | 42.9% | Nearly half viewed mentally ill persons as incapable of normal functioning. |
| Structural barriers | Inadequate workplace protections | 48.7% | Nearly half reported a lack of mental health accommodations at work. |
| | Systemic healthcare discrimination | 65.6% | Two-thirds experienced denial/delay of medical services due to a mental condition. |
| Misinformation | Fear of casual infection transmission | 41.7% | A significant proportion believed mental illness was contagious. |
| | Attribution to personal failure | 30.8% | One-third blamed individuals for their mental health condition. |
| Institutional factors | Perception of patient inferiority | 37.5% | Over one-third viewed the mentally ill as fundamentally different/inferior. |
| | Insufficient safety protocols | 34.1%* | One-third of healthcare workers reported inadequate protective measures in facilities. |

*Note*: * present (*n* = 87 healthcare workers).

**Table 4.** Attitudes of stigmatized patients toward help-seeking for mental health problems

| Attitude dimension | Positive response % (*n*) | Negative response % (*n*) | Neutral/ undecided % (*n*) |
|---|---|---|---|
| Help-seeking during crisis | 60.6% (254) | 24.1% (101) | 15.3% (64) |
| Belief in psychotherapy | 47.5% (199) | 26.1% (109) | 26.5% (111) |
| Willingness for prolonged care | 46.7% (196) | 29.6% (124) | 23.6% (99) |
| Perceived effectiveness of counseling | 41.5% (174) | 52.5% (220) | 6.0% (25) |
| Help-seeking as a last resort | 25.8% (108) | 54.9% (230) | 19.3% (81) |
| Preference for self-management | 21.7% (91) | 56.3% (236) | 22.0% (92) |

stigma dimensions reduced the odds of seeking care by 29–52%, supports Yu et al.'s (2023) model. This pattern is corroborated by similar findings in Sierra Leone, where individuals experiencing multiple stigma types were least likely to seek care (Betancourt et al., 2020; Yu et al., 2023). The specific influence of family exclusion as the strongest social factor (OR = 0.53) reflects a broader pattern in collectivist cultures, where familial disapproval often outweighs individual preferences, a trend also observed in studies from India (Mahomed et al., 2019). While our findings agree with broader patterns, they also reveal critical points of divergence, primarily related to the unique rural context of our study. The severity of economic stigma we observed exceeds the prevalence reported in urban Ghana (Daliri et al., 2024), suggesting that rural communities may experience compounded marginalization. These patterns of heightened rural stigma are not unique to Ghana, as similar geographic disparities have been documented in Kenya

(Mutiso et al., 2017), but the degree of difference highlights a significant urban–rural divide within the country. This urban–rural contrast is also evident in the social dimension. The persistently high prevalence of verbal abuse and family exclusion in our study stands in stark contrast to urban research in Ghana (Amadu and Hoedoafia, 2024), where community education programs have slightly decreased stigma related to mental illness. This disagreement with urban-focused findings underscores that stigma is not a monolith and that interventions successful in cities may not translate directly to rural settings with stronger kinship ties and different social dynamics, a nuance supported by evidence from Tanzania (Mascayano et al., 2015).

A notable area of partial agreement and nuance lies in structural stigma. The discrepancy between high educational exclusion (77.5%) and relatively lower healthcare discrimination (49.6%) partially aligns with analyses of Ghana's Mental Health Act (2012), which mandates healthcare access but lacks comparable protections in education (Walker and Osei, 2017). However, our data challenge any assumption of policy effectiveness; while systemic barriers in Ghana's healthcare system appear lower than in Nigeria (Chukwuma et al., 2024), the persistence of significant reported discrimination (65.6%) indicates that, as observed in Malawi, the existence of a legal framework does not ensure implementation in rural areas where enforcement is weak (Kainja, 2023).

Finally, the drivers of stigma related to mental health conditions in our study also show both alignment and deviation. The high endorsement of mental illness being contagious (41.7%) echoes trends found in other countries like Vietnam (Thi et al., 2024). However, the higher prevalence of these beliefs compared to urban Ghana (Daliri et al., 2024) suggests a critical point of disagreement regarding the penetration of mental health literacy, pointing to a specific informational deficit in rural areas that requires tailored, community-based education campaigns.

## Implications for policy and practice

The study highlights the urgent need for comprehensive interventions to combat stigma related to mental health conditions in rural Ghana. Policymakers should focus on enforcing anti-discrimination laws, especially in workplaces and educational settings, to reduce

**Table 5.** Multinomial regression analysis of mental health-related stigma and help-seeking behavior ($N = 419$)

| Stigma dimension | Specific form | Odds ratio (OR) | 95% CI | *p*-value | % Reduction in help-seeking |
|---|---|---|---|---|---|
| Economic stigma | Overall | 0.62 | [0.51–0.76] | 0.003 | 38% |
| | Employment discrimination | 0.62 | [0.51–0.75] | 0.003 | 38% |
| | Financial barriers | 0.59 | [0.47–0.73] | 0.001 | 41% |
| Social stigma | Verbal abuse | 0.71 | [0.58–0.87] | 0.008 | 29% |
| | Family exclusion | 0.53 | [0.42–0.67] | <0.001 | 47% |
| Psychological stigma | Internalized shame | 0.53 | [0.41–0.68] | <0.001 | 47% |
| Structural stigma | Healthcare system barriers | 0.48 | [0.37–0.62] | <0.001 | 52% |
| Interaction effects | Economic × Psychological | 0.41 | [0.32–0.53] | <0.001 | 59% |

Model fit statistics: $\chi^2 = 42.36$, df = 7, $p < 0.001$, Nagelkerke $R^2 = 0.31$.

economic and structural stigma. Community-based mental health literacy initiatives are vital to dispel misconceptions and lessen socio-cultural stigma. Engaging faith-based organizations and families in anti-stigma campaigns can harness local trust networks. Healthcare systems must integrate mental health services into primary care and train providers to address and prevent discrimination. Proactive outreach efforts should target individuals feeling internalized shame to promote early help-seeking.

## Limitations and future research

The study's cross-sectional design restricts causal inferences, and self-reported data may introduce bias. Focusing on a single rural district could limit the applicability of findings to urban or other regional settings, and the combination of patients, caregivers and healthcare workers, while providing a multifaceted perspective, may obscure group-specific experiences. Future research should adopt longitudinal approaches to examine stigma dynamics over time and incorporate qualitative methods to understand nuanced cultural perspectives. Investigation into the effectiveness of targeted interventions (e.g., family-centered programs or policy enforcement mechanisms) is necessary. Broader studies across various Ghanaian regions could reveal context-specific barriers and solutions.

## Conclusion

This study highlights how stigma related to mental illness functions as a complex barrier in rural Ghana, with socio-cultural, economic and structural factors intersecting to discourage help-seeking. While consistent with global patterns, the intensity of certain stigma forms calls for context-specific solutions. By addressing both individual perceptions and systemic inequalities, Ghana can make significant strides toward promoting mental health and well-being for all its citizens in line with Sustainable Development Goal 3.4, which aims at promoting mental health and well-being. The inclusion of new evidence from similar settings enhances the discussion, offering a comprehensive and nuanced understanding of stigma's effect on help-seeking behavior.

**Open peer review.** To view the open peer review materials for this article, please visit http://doi.org/10.1017/gmh.2025.10118.

**Supplementary material.** The supplementary material for this article can be found at http://doi.org/10.1017/gmh.2025.10118.

**Data availability statement.** The dataset for this study is available from the corresponding author upon reasonable request.

**Acknowledgments.** The authors thank the participants at the selected district facilities who took part in this research, without whom this work would not have been possible. We also appreciate the Faculty of Health and Allied Sciences at the Catholic University College of Ghana for the academic support provided to the first author in pursuing his degree program.

**Authors' contributions.** EKNB, EKA and EK conceptualized the study. EKA, SEA and EK developed the data collection instrument. SEA and EK supervised data collection. EKNB and PWA conducted the interviews. SEA and EK guided the entire data analysis and interpretation of the results. The manuscript was drafted by EKNB, EKA, GO and JAA, and critically revised by EKA, SEA and EK. All authors read and approved the final manuscript.

**Financial support.** This research received no specific grant from any funding agency, commercial or not-for-profit sectors.

**Competing interests.** The authors declare none.

**Ethics statement.** Before data collection, our project was reviewed and approved by the Ghana Health Service Ethics Review Committee (GHS-ERC 040/06/21). All participants were informed of the objective, justification and purpose of the study, and signed an informed consent form.

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
