## [Reviewer Report]

Effect of Mental Health Stigma on Help-Seeking Behaviour: A Community-Based Study in Sekyere South District, Ashanti Region, Ghana

This paper provides a very interesting investigation into the forms, drivers, and consequences of mental health-related stigma on help-seeking behaviour in the rural areas of Sekyere South District of Ghana. It puts forward an interesting discourse on the complex, multi-layered barrier, driven by socio-cultural beliefs and structural failures, necessitates urgent, multifaceted interventions targeting public education, policy, and self-stigma to improve mental health equity in rural Ghana. The paper finds that stigma considerably lowers help-seeking, with financial exclusion and internalized shame being the strongest restraints. This is an interesting paper, and it has the potential to make a strong contribution to the relevant bodies of literature. However, in its present form, it is underdeveloped, unsatisfactorily structured and not very well presented or written. The challenges for this paper are the following:

Title: I note that the title of your paper reads: Are Borders Only Physical? Intercultural Communication, Perceptions and Immigrant Experiences. It is better read as: Effects of Stigma on Help-Seeking Behaviour in Mental Health: A Community-Based Study in Ghana’s Sekyere South District in the Ashanti Region.

Abstract:

It would be useful if your abstract also contains the following.

□ In your opening statement in your abstract, in identifying the gap(s), it would be useful to put forward a meaningful statement. For example, stigma in mental health help-seeking acts as a critical barrier to care by fostering shame, fear of judgment, and discrimination, which deters individuals from seeking help, delaying treatment, and worsening outcomes, yet the area remains under-researched in low-resource context such as Ghana.

□ The aim of your study better reads as: ‘This study aimed to investigate the forms, drivers, and consequences of mental health-related stigma on help-seeking behaviour in the rural areas of Sekyere South District of Ghana’.

□ The type of literature that the paper drew on.

□ The average age of participants.

□ The method of data analysis, e.g. framework approach; thematic analysis.

Introduction

□ It would be useful if you can explain the opening statement a bit more. For example, mental health stigma creates a significant global barrier to accessing care, leading to delays in treatment, worsened symptoms, social exclusion, and poor health outcomes. This stigma stems from misunderstanding and prejudice, causing people to fear being labelled, ostracized, or judged, which diminishes self-esteem and hinders their willingness to seek help.

□ In your introduction, on lines 15-16, you need to explain the statement further to help your readers understand what you want to articulate. Do not assume, your readers understand issues around your topic. For example, you could explain that recent studies show stigma operates on public (others’ attitudes), self (internalized shame), and structural (discriminatory policies) levels, creating significant barriers to care by deterring people from seeking help for mental health conditions. This multi-level stigma can lead to social exclusion, shame, and inadequate access to services, negatively impacting treatment-seeking behaviours and health outcomes.

□ The gaps in literature would benefit from a clarification of the issue. The problematisation of your paper is needed much earlier in the section. The section is there to set up your paper, and let the reader know what the problem is, what we know about it, what we do not know about it (and thus, what your paper contributes to our knowledge). When doing this, be careful that the problematisation of the issue you are investigating is clear. In the current paper, the discussion around the investigation into the forms, drivers, and consequences of mental health-related stigma on help-seeking behaviour in the rural areas of Sekyere South District of Ghana is unfocussed.

□ For example, existing literature suggest that structural stigma and workplace discrimination are underexplored in low-resource context such as Ghana highlighting significant gaps in understanding these dynamics, particularly concerning marginalised groups such as persons living with mental health illnesses. In Ghana, research demonstrates the presence and severe impact of stigma across various sectors, including healthcare and employment, and underscores the need for context-specific, inclusive research to address these interconnected issues and dismantle exclusionary structures (You must support these with the relevant references).

Language/writing:

The writing needs improvement in places throughout the paper. I will suggest you write short sentences to improve meaning and clarity. Some of your sentences should be split into two.

□ In a formal journal article such as this, it would be useful to use formal phrases. For example, use ‘such as’ rather than ‘like’. Further, once you use the verb ‘like’ throughout your work, reviewers may think you wrote your work with AI.

Methods

□ You need to introduce ‘methods’ before moving on to ‘study design and setting.’ For example, methods examine study design and setting ………………………

□ You should clearly present the methods and research design of the study, detailing what was done, how it was done, and why it was done and describes the study context.

□ How do you address the issue of validity and reliability? How was your data triangulated?

□ What exactly did you do in relation to your research design, which involved quantitative methods.

□ You need to describe or give a profile of the participants and why each of them was chosen for the study.

□ What were the questions asked in your survey (structured questionnaires)? You can put this in the appendix, but you need to at least give an overview of your survey in the research methods?

Discussions

□ In the discussion compare the findings with what literature says- what are the points of agreement? and disagreement?

References

□ Cross-check your references for consistency.

---

## [Reviewer Report]

Thank you for the opportunity to review this interesting and important paper. It focuses on an under-researched topic and provides evidence from a low-resource setting on how multiple aspects of stigma play out.

Title: The title of the paper focuses on the relationship between stigma and help-seeking. While this is a component of the analysis, the results are more far reaching than this. In the purpose statement, for example, the authors indicate that they assess how stigma operates across social, economic, and healthcare systems. Perhaps the title could be broadened to reflect this.

Sample: The sample was purposive and included patients, caregivers of patients, and healthcare workers. I’m unclear as to why they were combined, given that they could experience stigma differently. For example, is internalized shame relevant for healthcare providers? I would expect healthcare providers to have different views about the efficacy of treatments. Were any stratified analyses conducted to check? Perhaps the authors could elaborate on the rationale for combining potentially disparate groups and identify this as a study limitation. Also, given that the sample was purposive, another limitation would be that it may not be representative of the underlying population. The high prevalences of stigma reported by participants may be due to sampling bias—a focus on those most likely to be at risk of being stigmatized.

Ethics: The subjects were told that they had the right to withdraw from the study at any time, but I wonder if they would have been able to withdraw their data once collected and anonymized. Perhaps a minor clarification would be helpful.

Results: Regarding the occupational categories almost half were “traders”. I am unclear as to what this is. Perhaps the authors could elaborate.

Conclusion: The conclusion references SDG 3.4. It is the first time it is referenced in the paper. Will readers know what this is?

Small Technical Issues:

Although often phrased as “mental health stigma” in fact, mental health is not stigmatized—mental illness or mental distress is. Perhaps, given the focus on patients, it could be rephrased to “mental illness related stigma”.

On page 11, line 30/31, the authors use the term “prevalence rates”. Technically, prevalences are proportions, not rates. I would suggest that the term “rate” not be used in this paper, given the cross-sectional nature of the data.